# First Study on the Electronic and Donor Atom Properties of the Ultra-Thin Nanoflakes Quantum Dots

**DOI:** 10.3390/nano12060966

**Published:** 2022-03-15

**Authors:** Laaziz Belamkadem, Omar Mommadi, Reda Boussetta, Mohamed Chnafi, Juán A. Vinasco, David Laroze, Laura M. Pérez, Abdelaziz El Moussaouy, Yahya M. Meziani, Esin Kasapoglu, Viktor Tulupenko, Carlos A. Duque

**Affiliations:** 1OAPM Group, Laboratory of Materials, Waves, Energy and Environment, Department of Physics, Faculty of Sciences, University Mohamed I, Oujda 60000, Morocco; belamkademlaaziz1905@gmail.com (L.B.); boussettareda7@gmail.com (R.B.); chnafimohamed5@gmail.com (M.C.); azize10@yahoo.fr (A.E.M.); 2Laboratory of Innovation in Science, Technology and Education, Regional Centre for the Professions of Education and Training, Oujda 60000, Morocco; 3Instituto de Alta Investigación, CEDENNA, Universidad de Tarapacá, Casilla 7D, Arica 1000000, Chile; juan.vinascos@udea.edu.co (J.A.V.); david.laroze@gmail.com (D.L.); 4Departamento de Física, FACI, Universidad de Tarapacá, Casilla 7D, Arica 1000000, Chile; lperez@uta.cl; 5Group of Nanotechnology, USAL-NANOLAB, Universidad de Salamanca, 37008 Salamanca, Spain; meziani@usal.es; 6Faculty of Science, Department of Physics, Sivas Cumhuriyet University, Sivas 58140, Turkey; ekasap@cumhuriyet.edu.tr; 7Donbass State Engineering Academy, 84313 Kramatorsk, Ukraine; viktor.tulupenko@gmail.com; 8Grupo de Materia Condensada-UdeA, Facultad de Ciencias Exactas y Naturales, Instituto de Física, Universidad de Antioquia UdeA, Calle 70 No. 52-21, Medellín AA 1226, Colombia; carlos.duque1@udea.edu.co

**Keywords:** ultra-thin quantum dot, nanoflakes, donor impurity, ground state energy, binding energy

## Abstract

Nanoflakes ultra-thin quantum dots are theoretically studied as innovative nanomaterials delivering outstanding results in various high fields. In this work, we investigated the surface properties of an electron confined in spherical ultra-thin quantum dots in the presence of an on-center or off-center donor impurity. Thus, we have developed a novel model that leads us to investigate the different nanoflake geometries by changing the spherical nanoflake coordinates (*R*, α, ϕ). Under the infinite confinement potential model, the study of these nanostructures is performed within the effective mass and parabolic band approximations. The resolution of the Schrödinger equation is accomplished by the finite difference method, which allows obtaining the eigenvalues and wave functions for an electron confined in the nanoflakes surface. Through the donor and electron energies, the transport, optoelectronic, and surface properties of the nanostructures were fully discussed according to their practical significance. Our findings demonstrated that these energies are more significant in the small nanoflakes area by altering the radius and the polar and azimuthal angles. The important finding shows that the ground state binding energy depends strongly on the geometry of the nanoflakes, despite having the same surface. Another interesting result is that the presence of the off-center shallow donor impurity permits controlling the binding energy, which leads to adjusting the immense behavior of the curved surface nanostructures.

## 1. Introduction

Recently, among the nanostructures that attract the attention of researchers, we find the ultra-thin quantum dots (QDs) on a nanometric scale due to their inherited characteristics and structural shape, which influences their electronic, optical, catalytic, and photoelectrochemical properties [1,2,3,4,5,6,7]. As it is known, the electron or hole confinement in 2D-QD or 3D-QD leads to augmenting the separation between energy levels, which produce a discrete energy spectrum with the decrease in nanostructures size. Along these lines, the confinement effect has been exploited to improve quantum information transport, sensor performance, and optoelectronic devices [8,9,10,11]. QDs and quantum wells have been used to build advanced components for electronic and optoelectronic applications [12,13,14]. In recent years, the 2D-QD semiconductors have been used as nanoflake photocatalysts due to the improvement of the lifetime, the rate of the adsorption, and desorption of the charge carriers confined at the nanostructures surface area [15,16,17]. Due to their importance, the electronic structure and bandgap, $E_{g}$, of GaSe solid solution nanoflakes are studied by using the density functional theory calculations [17]. The investigation of the $E_{g}$ value and the electronic structure shows that the semiconductor nanoflakes can absorb most of the solar radiation. On the other hand, the Si nanoflakes with a small thickness (15–17 nm) and large diameter (0.2–1.0 μm) obtained through the bead milling process are used to improve and recover sawdust as a high-performance negative electrode material for lithium-ion batteries [18].

Recent innovations and advancements in nanoscale technology, such as molecular beam epitaxy and metalorganic chemical vapor deposition or self-assembled technique, have created an opportunity to fabricate semiconductor quantum wells, quantum-well wires, and QDs in a variety of sizes and shapes, with applications spanning a wide range of fields. This allows new theoretical investigations to describe the phenomena associated with the low-dimensional nanostructures. Among these confinement system sets, we find the QDs, which have sparked much attention in recent years due to their potential uses in optoelectronic devices in order to produce a single photon and an associated photon pair [19,20]. The self-assembled approach is used to create bi-dimensional thin 2D-QD nanostructures such as disc, flat, and curved (nanoflake) surfaces with excellent quality and control throughout the growing process; the lateral diameters are significantly bigger than their thickness.

The confinement of charge carriers into the QD, according to two or three spatial directions, has many advantages in terms of control of their electronic and binding energies as well as their linear and nonlinear optical properties. It allows for better optimization of photophysical properties according to the shape and the QD size [21,22,23,24,25]. Due to the symmetric wave function, under linear and circularly polarized incident radiation, when the QD size changes under the condition of fixed geometry, small changes in the amplitude of the resonance peak and the progressive redshift of the curve can be identified. This behavior can be experimentally important in materials science in the coming years. In the past thirty years, scientists have paid considerable attention to the shape of the nanostructures, which is one of the most important factors to study the different properties of QD. In-depth researchers have been interested in spherical, cylindrical, and core/shell QD under external perturbation such as hydrostatic pressure, temperature, and electric and magnetic field [26,27,28,29,30,31,32,33]. It is important to note that the optical properties associated with off-center hydrogenic impurity states confined in cylindrical QD with various electric and magnetic fields directions have been reported by Heyn and Duque [30]. In this report, the authors show that the contribution of an off-donor impurity leads to extremely complex energy spectra, reduced squared dipole matrix elements, and optical absorption coefficients. These symmetry coefficients and phenomenology cannot be predicted. Therefore, numerical calculations have been proved to clarify these physical phenomena. Additionally, Shi et al. [31] have investigated the elliptic cylindrical core/shell QD and the non-axial electric field effects on the binding energy of the on-center hydrogenic impurity. They have found that the binding energy of cylindrical shape is higher than in elliptic cylinder shape. More recently, the θ-azimuthal angle effect, which produces systems of different geometry, on the electronic and optical properties of cylindrical and spherical nanostructures become the subject of many studies [32,33]. For the θ-azimuthal angle effect of cylindrical QD, we have recently obtained that [32]: (*i*) the optical bandgap depends strongly on the QD radius and its azimuthal angle, (*ii*) the geometric parameters (R,θ,H), the presence of impurity, and their positions in the rectangular base surface have a very important effect on the binding energy, (*iii*) the binding energy takes a critical value when the angle of the dot is almost equal to $40^{\circ }$ and takes the values of a 2D-QD when the opening angle tends towards $0^{\circ }$. Instead, the conduction band energy state properties of an electron interacting with donor impurity centers in a spherical segment (conical) GaAs/Al${}_{0.3}$Ga${}_{0.7}$As QDs have been theoretically studied [33]. This work shows that the red and blue mixed change in the optical absorption coefficient can be detected with the displacement of the impurities along the cone axis for a given cone size.

After we clarify low-dimensional nanostructures (the roles of the shape and size in 3D-QD) and their involvement in nanotechnology science, it is relevant to note that the 2D-QD ultra-thin nanostructures have some characteristics and properties that 3D-QD cannot provide. When controlling the confinement potential or the dot size for a fixed value between them, the charge carriers go from a volume state to a surface trapped state because the surface polarization produces an attraction of the carriers, trapped inside the QD, to be spontaneously trapped on the dot surface. The combined effects of the surface polarization instabilities of an exciton in a spherical semiconductor QD and the repulsive quantum confinement potential that surrounds it are studied in Ref. [7]. The interaction between these effects leads to confining the electron–hole pair inside the QD or near the surface when it is excited. Consequently, the QD surfaces, particularly the ultra-thin 2D-QD, have recently become important, resulting from the larger surface/volume ratio suitable for these nanostructures. We recall that the exact solution to the Schrödinger equation problem of an electron–hole pair confined on a microsphere (surface of a sphere), interacting via the Coulomb potential, with vanishing total angular momentum has been studied [34]. The motion state of Wannier exciton confined in the surface of the microsphere characterized by the radius *R* is clarified in Ref. [35]. Instead, Pramjorn and Amthong [36] have investigated the electric field effect on the off-center donor impurity states located in a curved two-dimensional InGaAs/GaAs with constant curvature using the finite difference method (FDM). They have found that the system size, curved plane radius, electric field direction, and impurity position significantly influence the binding energy. The control of these states, ground and first excited states of confined dopant atoms, which are regarded as logical states or quantum bits, leads to improved processing and storing quantum information. On the other hand, the confinement study of an off-center donor in semiconductor QD plays a key role in tailoring quantum devices.

Until now, there are no known theoretical or experimental reports about the impurity states confined in ultra-thin nanoflakes derived from a spherical surface QD. Consequently, we are interested in studying the electronic properties of an off-center donor confined in different ultra-thin nanoflakes geometry for further applications. In this work, we have investigated the donor binding energy related to the electronic and impurity states in the different geometries of ultra-thin 2D-QD, called nanoflake, made from GaAs. The different geometries of nanoflake derived from a spherical surface QD are characterized by the angles α (polar) and ϕ (azimuthal) and its radius. We consider the presence of a single donor and an electron, and the interaction between them through the Coulomb potential; they are confined at the same GaAs surface. After obtaining the Hamiltonian of spherical surface QD with infinite confining potential, the eigenvalues are obtained using the FDM to solve the two-dimensional effective mass partial differential equation. The dependence of the ground state electronic, donor, and binding energies by the spherical coordinates and many different donor positions are explored. The organization of this paper is as follows: Section 2 contains the theoretical framework, Section 3 discusses the nanoflakes energies results with and without an off-center donor; finally, we report our conclusions in Section 4.

## 2. Theoretical Framework

Let us consider a GaAs thinnest spherical QD, characterized by atomic-scale thickness, defined by the spherical coordinates (R,α,ϕ), with 0≤α≤π and 0≤ϕ≤2π, which makes it possible to produce different shapes of ultra-thin nanostructures named nanoflakes with the variation of spherical parameters (see Figure 1). The independent change of the α-polar and the ϕ-azimuthal angles, respectively, change the spherical surface to a moon surface and an umbrella surface for the complete values of the angles between them. Additionally, the curvature of nanoflakes can be controlled by spherical surface radius. We assume that the ionized donor atom is located at the atomic surface, which has a confinement potential equal to zero, embedded in a matrix of materials with considered infinite potential confinement. Within the effective mass approximation, in the presence of the donor impurity, the Hamiltonian of an electron confined in the spherical surface takes the form:(1)$$\begin{array}{cc} H_{D}= & -\frac{\hbar^{2}}{2m_{e}^{*}}\nabla^{2}+V_{D}\left(\theta ,\varphi \right)+V_{C}\left(\theta ,\varphi \right)\;, \end{array}$$
where $m_{e}^{*}$ is the electron effective mass, with $m_{0}$ corresponding to the free electron mass. Regarding the Equation (Equation 1), the electron kinetic energy operator is represented by the first term, and the second term is the Coulomb interaction between the donor impurity (with $r_{i}=R$, $\alpha_{i}$, and $\phi_{i}$ coordinates) and the electron (with r=R, θ, and φ coordinates), which is given by $V_{D}\left(\theta ,\varphi \right)=-\frac{e^{2}}{4\;\pi \;\varepsilon_{0}\;\varepsilon_{r}\;r_{ie}}$. Here, $\varepsilon_{0}$ is the vacuum permittivity, $\varepsilon_{r}$ is the GaAs static dielectric constant, and $r_{ie}$ is the electron-impurity distance, which can be approximated by $r_{ie}=\left|\vec{r_{i}}-\vec{r}\right|=\sqrt{2\;R^{2}\left[1-\cos\theta \;\cos\alpha_{i}-\sin\theta \;\sin\alpha_{i}\;\cos\left(\varphi -\phi_{i}\right)\right]}$, where φ(θ) and $\phi_{i}\left(\alpha_{i}\right)$ are, respectively, the azimuthal (polar) angular positions of electron and the impurity. The last term of Equation (Equation 1) is described by the quantum confinement potential of the surface system, which is defined as:(2)$$V_{C}\left(\theta ,\varphi \right)=\left\{\begin{array}{c} 0,\;\mathrm{if}\;0\leq \theta \leq \alpha \;\mathrm{and}\;0\leq \varphi \leq \phi \\ \infty ,\;\mathrm{otherwise}. \end{array}\right.$$

Outside the spherical QD surface, the probability density of the electron is zero. This means that the ground state wave function is also equal to zero. Inside the spherical QD surface, the eigenvalue time-independent Schrödinger equation without impurity, which gives the ground state electronic energy, $E_{0}$, and the corresponding wave function, $\psi_{0}\left(\theta ,\varphi \right)$, is written as:(3)$$\begin{array}{c} \left[-\frac{\hbar^{2}}{2m_{e}^{*}}\left(\frac{1}{R^{2}\;\sin\theta }\frac{\partial }{\partial \theta }\left(\sin\theta \frac{\partial }{\partial \theta }\right)+\frac{1}{R^{2}\;\sin^{2}\theta }\frac{\partial^{2}}{\partial \varphi^{2}}\right)+V_{C}\left(\theta ,\varphi \right)\right]\psi_{0}\left(\theta ,\varphi \right)=E_{0}\;\psi_{0}\left(\theta ,\varphi \right)\;. \end{array}$$

In the presence of a donor impurity, the following Schrödinger equation can be written:(4)$$\begin{array}{c} \left[-\frac{\hbar^{2}}{2\;m_{e}^{*}}\left(\frac{1}{R^{2}\;\sin\theta }\frac{\partial }{\partial \theta }\left(\sin\theta \frac{\partial }{\partial \theta }\right)+\frac{1}{R^{2}\;\sin^{2}\theta }\frac{\partial^{2}}{\partial \varphi^{2}}\right)+V_{D}\left(\theta ,\varphi \right)+V_{C}\left(\theta ,\varphi \right)\right]\psi_{D}\left(\theta ,\varphi \right)=E_{D}\;\psi_{D}\left(\theta ,\varphi \right)\;, \end{array}$$
where $E_{D}$ and $\psi_{D}\left(\theta ,\varphi \right)$, respectively, are the eigenvalue and its corresponding eigenfunction of a single donor confined in spherical QD surface.

The choice of spherical coordinates allows obtaining different geometries associated with the change in azimuthal and polar angles with heterostructures ranging from spherical surfaces to moon and umbrella shapes. It is possible that with the variation of the radius, its curvature can be controlled. On the other hand, for example, cylindrical coordinates give only a geometry controlled by the curvature radius [36].

In order to solve the Schrödinger equations of the nanosystem, we have used the FDM to determine the eigensolutions of partial differential equations. In this computation, the Equation (Equation 4) of nanoflakes are discretized on a uniform two-dimensional mesh containing $N_{\theta }\times N_{\varphi }$ grid points according to finite difference technique. We divide the intervals $\left[0,\alpha \right]$ into $\left(N_{\theta +1}\right)$ and $\left[0,\phi \right]$ into $\left(N_{\varphi +1}\right)$ parts. We consider $\theta_{i}=i\Delta \theta$ and $\varphi_{j}=j\Delta \varphi$, where $1\leq i\leq N_{\theta }$ and $1\leq j\leq N_{\varphi }$ are, respectively, the integers identifying the positions of grid points along the nanoflake surface identified by the θ and φ angles, and the grids-spacings along the surface are presented by Δθ and Δφ. Using the second order central difference approximation for $\frac{\partial \psi_{D}\left(\theta ,\varphi \right)}{\partial \theta },\frac{\partial^{2}\psi_{D}\left(\theta ,\varphi \right)}{\partial \theta^{2}}$ and $\frac{\partial^{2}\psi_{D}\left(\theta ,\varphi \right)}{\partial \varphi^{2}}$, we obtain the discretized Equation (Equation 4) in a grid point (i,j) as
(5)$$\begin{array}{cc} \; & -\frac{\hbar^{2}}{2m_{e}^{*}}\left[\left(\frac{1}{R^{2}{\left(\Delta \theta \right)}^{2}}+\frac{1}{2}\frac{\cos(i\Delta \theta )}{R^{2}\sin\left(i\Delta \theta \right)\Delta \theta }\right)\psi_{D}^{i+1,j}-\left(\frac{2}{R^{2}{\left(\Delta \theta \right)}^{2}}+\frac{2}{R^{2}}\frac{1}{\sin^{2}\left(i\Delta \theta \right){\left(\Delta \varphi \right)}^{2}}+\frac{2m_{e}^{*}}{\hbar^{2}}\left(V_{D}^{i,j}-V_{C}^{i,j}\right)\right)\psi_{D}^{i,j}\right]\\ & +\frac{\hbar^{2}}{2m_{e}^{*}}\left[\left(\frac{-1}{R^{2}{\left(\Delta \theta \right)}^{2}}+\frac{1}{2}\frac{\cos(i\Delta \theta )}{R^{2}\sin\left(i\Delta \theta \right)\Delta \theta }\right)\psi_{D}^{i-1,j}+\frac{1}{R^{2}}\frac{\psi_{D}^{i,j-1}}{\sin^{2}\left(i\Delta \theta \right){\left(\Delta \varphi \right)}^{2}}+\frac{1}{R^{2}}\frac{\psi_{D}^{i,j+1}}{\sin^{2}\left(i\Delta \theta \right){\left(\Delta \varphi \right)}^{2}}\right]=E_{D}\psi_{D}^{i,j}\;\;, \end{array}$$
where $V_{C}^{i,j}\left(\theta ,\varphi \right)$$\left(V_{D}^{i,j}\left(\theta ,\varphi \right)=\frac{-e^{2}}{4\pi \varepsilon_{0}\varepsilon_{r}\sqrt{2R^{2}\left[1-\cos\left(i\Delta \theta \right)\cos\alpha_{i}-\sin\left(i\Delta \theta \right)\sin\alpha_{i}\cos\left(j\Delta \varphi -\varphi_{i}\right)\right]}}\right)$ is the confinement (Coulomb) potential at the (i,j)-point and $\Delta \theta =\frac{\alpha }{N_{\theta }+1}$$\left(\Delta \varphi =\frac{\phi }{N_{\varphi }+1}\right)$ is the pitch angle between two consecutive grid points in ${\hat{e}}_{\theta }\left({\hat{e}}_{\varphi }\right)$ direction.

In order to detail our calculation, we write the Equation (Equation 5) in a system of $N_{\theta }\times N_{\varphi }$, with respect the boundary conditions by the Dirichlet and Newman boundary, in the following matrix form:(6)$$\begin{array}{cc} \; & \left[\begin{array}{cccccccc} \beta_{11}-E_{D} & u_{11} & 0 & \cdots & \gamma_{11} & \cdots & \cdots & 0\\ w_{12} & \beta_{12}-E_{D} & u_{12} & 0 & \cdots & \gamma_{12} & \cdots & \cdots \\ 0 & w_{13} & \beta_{13}-E_{D} & u_{13} & 0 & \cdots & \gamma_{13} & \cdots \\ \vdots & 0 & w_{14} & \cdots & \cdots & \cdots & \cdots & \cdots \\ \delta_{21} & \cdots & \cdots & \cdots & \cdots & \cdots & \cdots & \cdots \\ 0 & \delta_{22} & \cdots & \cdots & \cdots & \cdots & \cdots & \cdots \\ \vdots & 0 & \delta_{23} & \cdots & \cdots & \cdots & \cdots & \cdots \\ \vdots & \cdots & 0 & \cdots & \cdots & \cdots & \cdots & \cdots \\ \vdots & \cdots & \cdots & \cdots & \cdots & \cdots & \cdots & \cdots \\ \vdots & \cdots & \cdots & \cdots & \cdots & \delta_{N_{\theta }N_{\varphi }-2} & \cdots & 0\\ \vdots & \cdots & \cdots & \cdots & \cdots & \cdots & \delta_{N_{\theta }N_{\varphi }-1} & \cdots \\ 0 & \cdots & \cdots & \cdots & \cdots & \cdots & \cdots & \delta_{N_{\theta }N_{\varphi }} \end{array}\right.\\ & \left.\begin{array}{cccc} \cdots & \cdots & \cdots & \vdots \\ \cdots & \cdots & \cdots & \vdots \\ \cdots & \cdots & \cdots & \vdots \\ \cdots & \cdots & \cdots & \vdots \\ \cdots & \cdots & \cdots & \vdots \\ \cdots & \gamma_{N_{\theta }N_{\varphi }-2} & \cdots & \vdots \\ \cdots & \cdots & \gamma_{N_{\theta }N_{\varphi }-1} & 0\\ \cdots & \cdots & \cdots & \gamma_{N_{\theta }N_{\varphi }}\\ \cdots & \cdots & \cdots & \vdots \\ w_{N_{\theta }N_{\varphi }-2} & \beta_{N_{\theta }N_{\varphi }-2}-E_{D} & u_{N_{\theta }N_{\varphi }-2} & 0\\ \cdots & w_{N_{\theta }N_{\varphi }-1} & \beta_{N_{\theta }N_{\varphi }-1}-E_{D} & u_{N_{\theta }N_{\varphi }-1}\\ \cdots & \cdots & w_{N_{\theta }N_{\varphi }} & \beta_{N_{\theta }N_{\varphi }}-E_{D} \end{array}\right]\left[\begin{array}{c} \psi_{D}^{1,1}\\ \psi_{D}^{1,2}\\ \psi_{D}^{1,3}\\ \vdots \\ \vdots \\ \vdots \\ \vdots \\ \vdots \\ \vdots \\ \psi_{D}^{N_{\theta },N_{\varphi }-2}\\ \psi_{D}^{N_{\theta },N_{\varphi }-1}\\ \psi_{D}^{N_{\theta },N_{\varphi }} \end{array}\right]=0\;\;, \end{array}$$
where $\beta_{i,j}=\frac{\hbar^{2}}{2m_{e}^{*}}\left[\frac{2}{R^{2}{\left(\Delta \theta \right)}^{2}}+\frac{2}{R^{2}}\frac{1}{\sin^{2}\left(i\Delta \theta \right){\left(\Delta \varphi \right)}^{2}}\right]+V_{D}^{i,j}-V_{C}^{i,j},$ $\gamma_{i,j}=\frac{-\hbar^{2}}{2m_{e}^{*}}\left[\frac{1}{R^{2}{\left(\Delta \theta \right)}^{2}}+\frac{1}{2R^{2}}\right.$

$\left.\frac{\cos(i\Delta \theta )}{\sin(i\Delta \theta )\Delta \theta }\right],$$\delta_{i,j}=\frac{-\hbar^{2}}{2m_{e}^{*}}\left[\frac{1}{R^{2}{\left(\Delta \theta \right)}^{2}}-\frac{1}{2R^{2}}\frac{\cos(i\Delta \theta )}{\sin(i\Delta \theta )\Delta \theta }\right],$ and $u_{i,j}=w_{i,j}=\frac{-\hbar^{2}}{2m_{e}^{*}}\left[\frac{1}{R^{2}}\frac{1}{\sin^{2}\left(i\Delta \theta \right){\left(\Delta \varphi \right)}^{2}}\right]$.

After solving numerically the matrix of $N_{\theta }^{2}\times N_{\varphi }^{2}$ dimension (Equation (Equation 6)) with $N_{\theta }\times N_{\varphi }$ unknowns, the donor eigenenergy and the eigenfunction $(E_{D}$ and $\psi_{D}\left(\theta ,\varphi \right))$ will be found. As well as that, eigenenergy ($E_{0}$) and eigenfunction ($\psi_{0}\left(\theta ,\varphi \right)$) of electron can be obtained by using the FDM. In the absence of impurity, the numerical results obtained are the same as the exact eigenenergies. In this work, we also examined the ground state donor binding energy, which is defined as:(7)$$E_{B}=E_{0}-E_{D}\;.$$

In the next section (Section 3), we will introduce our results through corresponding discussions, in which we analyze the influence of the nanoflakes geometry and impurity position on the behavior of the binding, electronic, and donor energies.

## 3. Results and Discussion

In our numerical computation, we have examined the ground state electronic, donor, and binding energies of an off-center impurity in different nanoflakes geometry derived from the GaAs thinnest spherical 2D-QD by fitting the spherical coordinates (R,α,ϕ), with 0≤α≤π and 0≤ϕ≤2π and adopting the model described above. We have named the spherical coordinates by R,α, and ϕ to not confuse them with the spatial coordinate of electron. The impurity is located along the main axes of the nanoflakes, i.e., the axes that divide the 2D surface into two equal areas. The values of electron effective mass $m_{e}^{*}\left(\mathrm{GaAs}\right)=0.067\;m_{0}$, where $m_{0}$ is the free electron mass, and the dielectric constant is $\varepsilon_{r}\;\left(\mathrm{GaAs}\right)=12.4$. To start the discussion of our numerical results, the FDM was adopted for N=60×60 nodes, and the surface of the nanoflakes QD equals $S_{nf}=R^{2}\times \phi \left(1-\cos\left(\alpha \right)\right)$, where ϕ and α are the maximum azimuthal and polar angles corresponding to the edges of the nanoflakes, respectively.

To study the surface electronic states of nine nanoflakes geometries (see Figure 2a), we have illustrated in Figure 2, the variation of the ground state electronic energy as a function of the radius (b), the ϕ-angle (c), and the α-angle (d) of spherical QD surface. The uncorrelated electron energy (without impurity) is obtained through solving the Schrödinger equation by omitting the Coulomb potential ($V_{D}=0$) and using the electronic wave function $\psi_{0}$ (see Equation (Equation 3)). In Figure 2b–d, we can observe that the electron energy keeps the same behavior as a function of the geometrical parameters, in which it increases with decreasing the nanoflakes surface. Due to the reduction in radius and angles (α and ϕ), a decrease occurs in the surface of the confinement area. For the maximum values of α and ϕ, i.e., $\alpha =180^{\circ }$ and $\phi =360^{\circ }$, the electron energy behavior of the system is similar to that of an ultra-thin spherical QD [35]. On the other hand, for a given value of $\phi \leq 360^{\circ }$ and $\alpha \leq 180^{\circ }$, the enhancement of the curvature radius produces nearly flat nanoflakes, and their electronic energies tend to the eigenenergies of a quantum well, which is plotted in Figure 2b. We note that the increase of the 2D-QD radius with $\phi \leq 360^{\circ }$ and $\alpha \leq 180^{\circ }$ decreases the confinement of the particles and increases their degree of freedom along the surface, and when the surface is thinner, the electronic energy tends towards their energy in a well-confined quantum well. Again, for a given radius value and infinite combinations of the ϕ and α-angles, we can get the same nanoflakes surface values ($S_{nf}$). However, our calculations show that the electron energy in two various structures with the same surface is different, which explains that the ground state electronic energy value of the nanoflakes shapes with $\phi =360^{\circ }$ and $\alpha =120^{\circ }$ and $\phi =270^{\circ }$ and $\alpha =180^{\circ }$ ($S_{nf}=942$ nm${}^{2}$) or $\phi =360^{\circ }$ and $\alpha =90^{\circ }$ and $\phi =180^{\circ }$ and $\alpha =180^{\circ }$ ($S_{nf}=628$ nm${}^{2}$) are different, due to the angular symmetry and the nature of system curvature. Consequently, the nanosystem shape produces an effect that can be added to the size effect [23]. The results show that the angular confinement of 2D nanosystems can significantly alter the conduction band energy caused by the high density and the strong quantum confinement of the electron, which produces a larger bandgap in the nanoflakes than the overall bandgap of the bulk material and other quantum systems. This behavior can be helpful for the possibility of enhancing the electric properties of optoelectronic devices by improving the ohmic contact of different thicknesses of nanoflakes [6].

For purposes of clarity, for the considered different setups of the structure angles and taking a fixed value of the radius of 10 nm, we list the relationship between angles and area of the nanofleck (ϕ, α, $S_{nf}$):

(360${}^{\circ }$, 180${}^{\circ }$, 1256 nm${}^{2}$), (360${}^{\circ }$, 120${}^{\circ }$, 942 nm${}^{2}$), (360${}^{\circ }$, 90${}^{\circ }$, 628 nm${}^{2}$), (360${}^{\circ }$, 45${}^{\circ }$, 184 nm${}^{2}$), (270${}^{\circ }$, 180${}^{\circ }$, 942 nm${}^{2}$), (180${}^{\circ }$, 180${}^{\circ }$, 628 nm${}^{2}$), (90${}^{\circ }$, 180${}^{\circ }$, 314 nm${}^{2}$), (45${}^{\circ }$, 180${}^{\circ }$, 157 nm${}^{2}$), (30${}^{\circ }$, 180${}^{\circ }$, 105 nm${}^{2}$).

In order to investigate the characteristics of quantum states, we continue to include the presence of an on-center donor impurity atom in this problem. We are interested in approaching the exact ground state of a 2D hydrogenic atom in different nanoflakes geometries. As we can remark in Figure 3, the donor state energy is highly sensitive to the ϕ and α pair and the *R*-radius. This figure shows that the variations of the donor energy curves are unstable affected by the radius for different α and ϕ angles due to the competition between the interaction types (attractive and repulsive). The interaction between the electron and the donor atom is described by the Coulomb potential, which strongly depends on the inter-particle $r_{ie}$-distance. If $r_{ie}$ is large, the permanent Van Der Waals attractive force will dominate, which explains the donor energy value close to −8 meV when the surface radius is more significant, and this energy in the ground state becomes bigger in spherical 2D-QD. On the other hand, when the surface radius diminishes, the probability density of electron approaches near the donor atom located at the center of the nanoflakes, which applies between them an attractive interaction responsible for the appearance of the minimum. This critical radius becomes more important, in which the donor atom is more stable when the nanoflakes reach the spherical surfaces due to the abrupt shift from the attractive to the repulsive forces of the donor in the case of spherical 2D-QD. For strong confinement, when the distance between particles becomes very small, the repulsive forces become more significant compared to the attractive forces and cause a rapid increase in the donor energy due to the repulsive forces, and quantify states can then be formed. In this confinement regime, the donor energy value in nanoflakes with small surfaces is more significant than that in large surfaces. On the right-hand side of the figure, the results of the normalized wavefunctions corresponding to the ground and the first three excited states for R=10 nm, $\alpha =180^{\circ }$, and $\phi =30^{\circ }$ are shown. We can notice that the wavefunctions vanish at the boundaries of the structure, which is imposed by the Dirichlet conditions in the problem. Due to the relatively small value of the ϕ-angle ($\phi =30^{\circ }$), it is clear that the excited states are fundamentally associated with the appearance of antinodes along the longest dimension of the structure.

To clearly understand the ϕ and α-angle effects, corresponding to different nanoflakes shapes, on the ground state donor atom, we have examined in Figure 4, the donor energy as a function of the ϕ-azimuthal (a) and α-polar angles (b) for a given value of curved radius (R=20 nm) keeping the impurity donor always localized at the nanoflake gravity center. As can be seen from Figure 4a, for some significant values of polar-angle, the donor energy rapidly diminishes as the azimuthal-angle enhances until it reaches a minimum, in which the atom will be more stable and then increases weakly. Indeed, the donor energy is higher when the confinement system is more significant. As the confinement angles are smaller, the donor atom energy is higher, which is more evident in Figure 4a,b. However, the numerical computations show that the donor energy is enhanced by increasing the polar angle for a large value of the azimuthal angle. This behavior can be clarified because the attractive Coulombic effect is more meaningful than the confinement one; consequently, the attractive force dominates. The results show that the variation of the donor atom energy versus the thinnest spherical radius and the ϕ and α-angles takes, respectively, the conductive bond and secondary binding types following the Lennard Jones curves. The combined variation of the angles and the curved radius of nanoflakes leads to adjusting the 2D hydrogen atom energy, and they can electronically saturate its atoms. As a result of the bonds that the donor atom links with the neighboring atoms of the same material produce an adsorption type on a surface by improving the thermodynamic properties of 2D nanoflakes, which provides new technological possibilities [19,37,38].

Intending to clarify our analysis, we examine in Figure 5, Figure 6 and Figure 7 the behavior of the electron that interacts with the donor impurity by specifically investigating the impact of the spherical surface coordinates and the impurity position on the binding energy. In Figure 5, we are interested in studying the size effect, adjusted by the curvature radius, for nine nanoflakes geometries with the impurity located at the gravity center of each nanoflake. According to the infinite confinement potential model adopted, the behavior of the binding energy follows a well-known tendency, which increases with the enhancement of the confinement effect. Similarly, the donor binding energy decreases monotonically with increasing *R*, which is linked to the augmentation in the effective $r_{ie}$ inter-particle distance due to the expansion of the nanoflakes area. As can be noticed along the interval of the considered radius, a slight increase of the binding energy was detected with the reduction of the polar (α) or azimuthal (ϕ) angles, mainly in the nanoflakes of low radius. The increase of the confinement effect (the 2D system tends to zero, i.e., α→0 and ϕ→0) produces a greater electron-donor spatial localization in a small region, and their kinetic energy becomes more important. Note that there are crossings between some binding energy curves (for example, R=15 nm, corresponding to the crossings between the moon and the umbrella surfaces), which is more important for the nanoflakes fabrication. For greater values of radius, R→∞, two confinement effects can build it: (*i*) for the nanoflake with $\alpha =45^{\circ }$ and $\phi =360^{\circ }$, its binding energy takes the ultra-thin 2D quantum disk value, and (*ii*) for $\phi =45^{\circ }$ and $\alpha =180^{\circ }$, the nanoflake tends to approach an ultra-thin flat eye form, and their binding energy is smaller than 2D quantum disk. Controlling the area of the nanoflake produces an energy shift of optical responses to high energy (blue shift) by decreasing the curved radius and the angles into it.

To give more information about the effects of the angles that describe the edges of the nanoflakes on the binding energy of a donor impurity placed at the center of nanostructure with a curved radius R=20 nm, we present in Figure 6 its variation as a function of the ϕ-polar angle, for several azimuthal angle values, Figure 6a, and as a function of the α-azimuthal angle, for several polar angle values, Figure 6b. In Figure 6a, the donor binding energy shows a monotonically decrease with augmenting the polar angle of the nanostructures. An increase of the azimuthal angle means an increase in dot area, so the orbital of a mono-donor is more extended, and consequently, the electron–donor attraction is reduced. We have plotted in Figure 6b the variation of the binding energy versus ϕ in order to visualize the critical value of the mono-donor binding energy when the system is more stable. For $\alpha =180^{\circ }$, when ϕ decreases from $360^{\circ }$ to zero, the binding energy increases and moves from the thinnest spherical surface towards a semicircle 1D quantum wire, and consequently, the electron wave function has *s*-like symmetry, which is distributed uniformly along with it. For $\alpha \leq 180^{\circ }$, a strong behavior of the binding energy appeared. Indeed, for a fixed polar angle, the binding energy is enhanced by diminishing the azimuthal angle until it reaches a critical value, with the maximum stability of the system that can be obtained and then diminishes. This effect becomes more significant for low ϕ value due to the nanoflakes shape, which does not keep the symmetrical distribution of the wave function.

The donor atom can be located anywhere on the nanoflakes, and its displacement in the different directions (${\hat{e}}_{\theta }$ and ${\hat{e}}_{\varphi }$) of quantum confinement systems is considered as a significant factor apropos the modification of the binding energy. To explain the displacement effect along with the main directions of each nanoflake shape, we have examined in Figure 7, the variation of the ground state impurity binding energy corresponding to the shapes shown in Figure 2a. As seen in Figure 7a–c, the binding energy takes the maximum value when the impurity is located at the gravity center of the nanostructures due to the maximum probability density of electrons in this position. Consequently, the electron–donor bound is more important. For each curve of the Figure 7a,b, it is found that the binding energy increases until it reaches the maximum value, then it decreases, when the displacement of the donor is from 0 to α according to the ${\hat{e}}_{\theta }$ direction along the edge of the nanoflakes ($\phi_{i}=\phi$). Consequently, the electron–donor interaction is less bound when the impurity is located at $\alpha_{i}\;=\;0^{\circ }$ and $\;180^{\circ }$. For a given radius value (R=10 nm and 20 nm), the figure shows that the effect of the ϕ-azimuthal angle on the binding energy is more significant when the impurity position is near the center ($\alpha_{i}\;=\;\alpha /2$). For this reason, we have concentrated on the polar position of the donor $\alpha_{i}\;=\;\alpha /2$ in the previous figures. Furthermore, we can see from the same figure that as the *R*-radius diminishes, the reduction of the azimuthal angle leads to augment the binding energy, and their effect is clearer for R=10 nm than for 20 nm, due to the substantial limitation of the electronic wave function. In contrast, the augmentation of the azimuthal angle enhances the binding energy when the impurity is near the corner of nanoflakes structures. This behavior can be described as follows: (*i*) the symmetry of the atomic orbitals is limited to the corner and (*ii*) the probability density spreads near from the donor position. The augmentation of the binding energy with the augmentation of the nanostructures aperture when the impurity is localized in the corner, which is discussed in the previous works [32,33]. We can also see from Figure 7b that the maximum value of the binding energy will be moved from the center ($\alpha_{i}\neq \;0.5\;\alpha$) when the polar angle is not complete $\alpha <180^{\circ }$, since the system does not keep its symmetry and the maximum value increases when the polar angle decreases.

For a complete value of the azimuthal angle, $\phi =360^{\circ }$, the variation of the binding energy as a function of polar impurity position for various polar angles is plotted in Figure 7c. From this figure, we can observe that the binding energy is maximum when the donor is located at the spherical cap ($\alpha_{i}=0^{\circ }$) and decreases monotonically when the three polar angles are not completed ($\alpha =45^{\circ },\alpha =90^{\circ }$, and $\alpha =120^{\circ }$). Owing to the weak localization of the electron wave function, the electron–donor Coulomb interaction decreases as the donor moves towards the edges, resulting in a decrease of the binding energy. It is important to notice that the binding energy takes the same value when $\alpha_{i}=0$ and $180^{\circ }$, a cause to the symmetry of the nanostructure for $\alpha =180^{\circ }$ and the ground state wave function, as well as the Coulombic potential of a donor, displace along towards the ${\hat{e}}_{\theta }$ direction.

In order to give more pieces of information about the displacement effect of the impurity along with the ${\hat{e}}_{\phi }$-direction, we have plotted in Figure 7d the variation of the binding energy as a function of the azimuthal position of the impurity for three azimuthal angles, $\phi =30^{\circ },45^{\circ }$, and $90^{\circ }$ with $\alpha_{i}=\alpha /2=90^{\circ }$. As the impurity position is allowed to shift from the first edge to the second edge of the nanoflake, the ground state binding energy initially increases, reaches a limit, and then decreases. It is more important in $\phi_{i}=0^{\circ }$ than in $\phi_{i}=\phi$. The binding energy maximum value is displaced as the impurity is shifted around $\phi_{i}=0.2\;\phi$. The different maximums can be explained by the fact that the impurity is positioned near to the gravity center and the interaction between electronic density with the donor Coulombic potential. It can also be shown that for smaller azimuthal angle ϕ=30, the binding energy is greater, and their maximum is equal to 65.30 meV for a single donor located at $\phi_{i}=7^{\circ }$ and $\alpha_{i}=90^{\circ }$ and confined in a nanoflake sized by R=10 nm, $\phi =30^{\circ }$, and $\alpha =90^{\circ }$.

## 4. Conclusions

This paper presents a theoretical study of several GaAs ultra-thin nanoflakes geometries by investigating their energetic characteristics. Indeed, we have specifically discussed our findings of the effects of the curvature, geometry, and size on the electronic, donor atom, and binding energies. Concave and flat nanoflakes 2D-QD controlled by the curvature radius were obtained. The nanoflakes structures derived from a spherical surface QD are characterized by respecting the parameters (the radius *R* and the two angles α and ϕ). With an infinite potential barrier, the ground state eigenenergy and wave function of an electron confined are determined from the resolution of the 2D Schrödinger equation within the effective mass approximation by using a two-dimensional finite difference method. The presence of an on-center or off-center donor impurity anywhere on the nanoflakes QD is considered. Without a donor impurity, the results reveal that the electron energy monotonically decreases by augmenting the nanoflakes parameters (*R*,α, and ϕ). We have shown that the electron energy value for the vertical hemispherical surface nanostructure is more pronounced than in horizontal hemispherical surface form one, although they have the same surface values. In the presence of the donor atom localized at the gravitational center of each nanoflake, for a smaller curvature radius, the donor energy presents an increasing behavior when the nanoflakes area becomes small, through decreasing the azimuthal and polar angles. The computations have also been attached to the binding energy for an on-center and off-center donor atom. We have found a slight increase of the binding energy with the reduction of the pair (α,ϕ) angles, mainly in the low radius nanoflakes, and a crossing between two binding energies of two different geometries has been detected. A strange behavior has been observed for the variation of the binding energy with the decrease of the polar angle, which explains the transition from a 2D system to a 1D system. The displacement of the donor impurity to the edges, specially in the main axes of the nanostructures, produces a maximum value of the binding energy, and the amplitude of this maximum increases with decreasing the nanoflakes area. In contrast, the impurity position has a high energy efficiency on surface states, which can improve the performance of optoelectronic devices based on very thin 2D nanoflakes. It should be underlined that this work at nanoscale shape constitutes the first theoretical study on the electronic properties related to the off-center donor impurity confined in different geometries of the ultra-thin nanoflakes. We have to stress that in our model, we have considered a surface with curvature that follows a spherical geometry, that is, our system consists of a surface taken as the portion of a sphere with zero thickness. In this sense, our model is ideal since in general the layered systems have at least the thickness of an atom. Additionally, in our model, we have considered an infinite confinement potential, which is an excellent model to describe nanoflakes surrounded by vacuum or air, which translates into an infinite confinement potential that prevents the charge carriers from escaping from the heterostructure to the surrounding atmosphere or vacuum region. We hope that our study will be helpful for further theoretical and experimental investigations in two-dimensional systems, especially in Van Der Waals systems, where many questions are still open.

## Figures and Tables

**Figure 1 nanomaterials-12-00966-f001:**
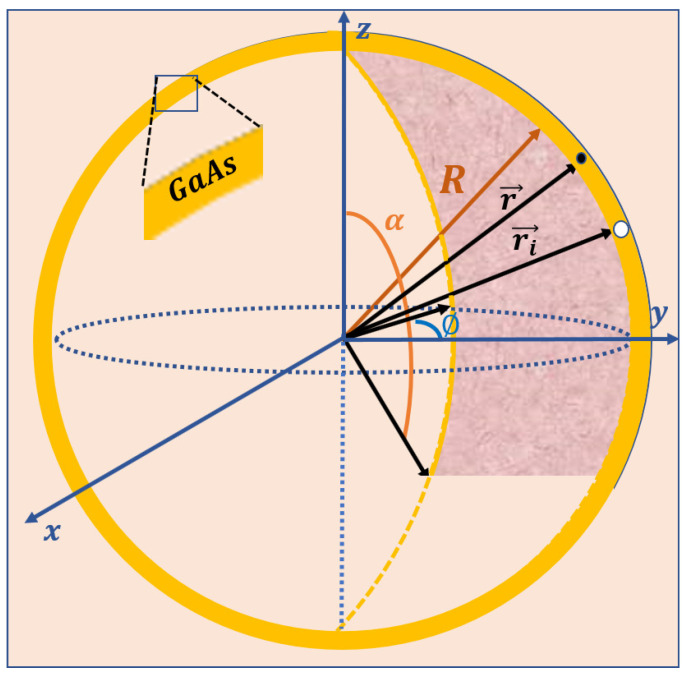
Illustration of the ultra-thin nanoflakes geometry derived from the spherical surface coordinates (*R*, α,ϕ). The positions of the electron and the donor atom have been, respectively, considered by $\vec{r}$ and $\vec{r_{i}}$.

**Figure 2 nanomaterials-12-00966-f002:**
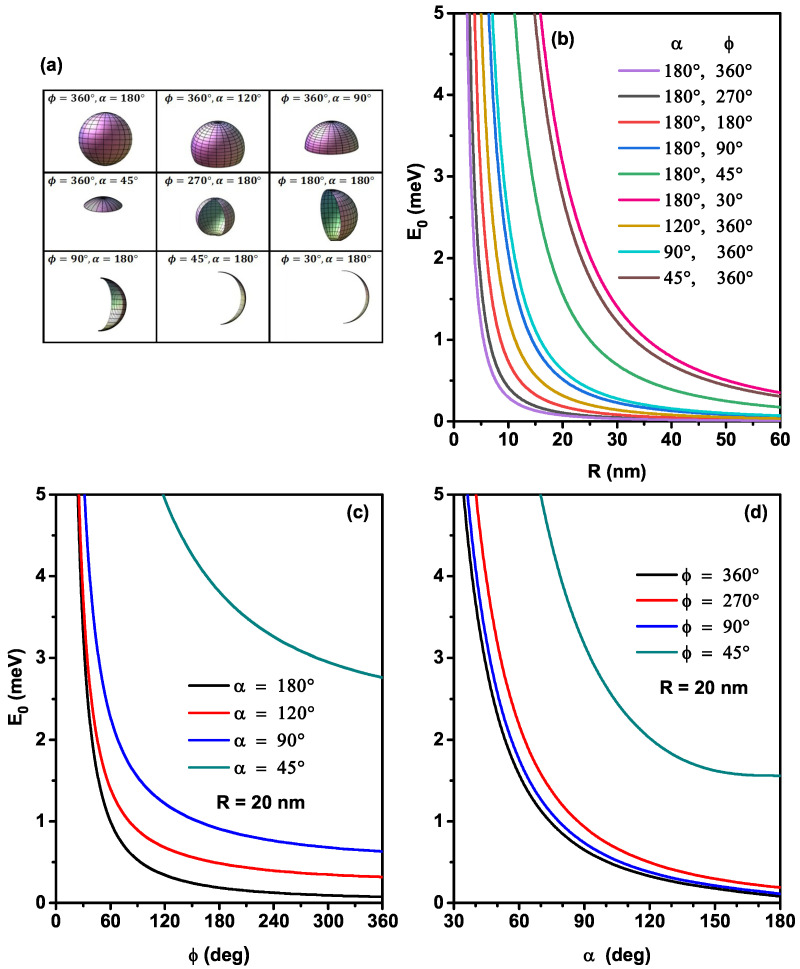
Schematic view of nine geometries of the ultra-thin nanoflakes 2D-QD considered in the present study (**a**). Electron energy as a function of the radius for nine nanoflakes shapes (**b**). The ϕ-angle dependence for the electron energy with several values of the α-angle with R=20 nm (**c**). The α-angle dependence of the electron energy for several values of the ϕ-angle with R=20 nm (**d**).

**Figure 3 nanomaterials-12-00966-f003:**
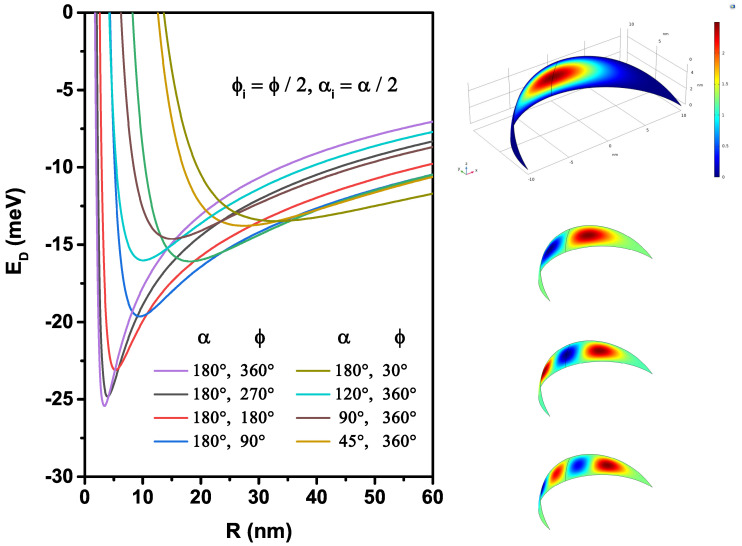
Variation of the ground state energy of an on-center donor impurity as a function of nanoflakes radius for different setups of the α and ϕ-angles. On the right side the results of the normalized wavefunctions corresponding to the ground and the first three excited states for R=10 nm, $\alpha =180^{\circ }$, and $\phi =30^{\circ }$ are shown. The red and blue colors correspond to the maximum and minimum of the wave function, respectively.

**Figure 4 nanomaterials-12-00966-f004:**
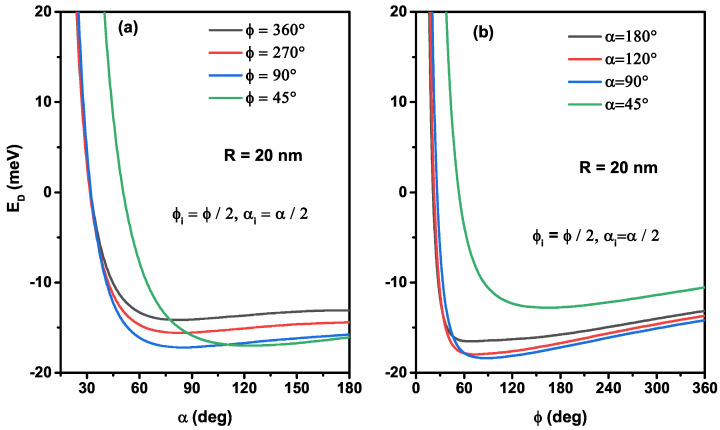
The on-center donor impurity energy as a function of the α-angle for several values of the ϕ-angle (**a**) and as a function of the ϕ-angle for several values of the α-angle (**b**). Calculations are for R=20 nm.

**Figure 5 nanomaterials-12-00966-f005:**
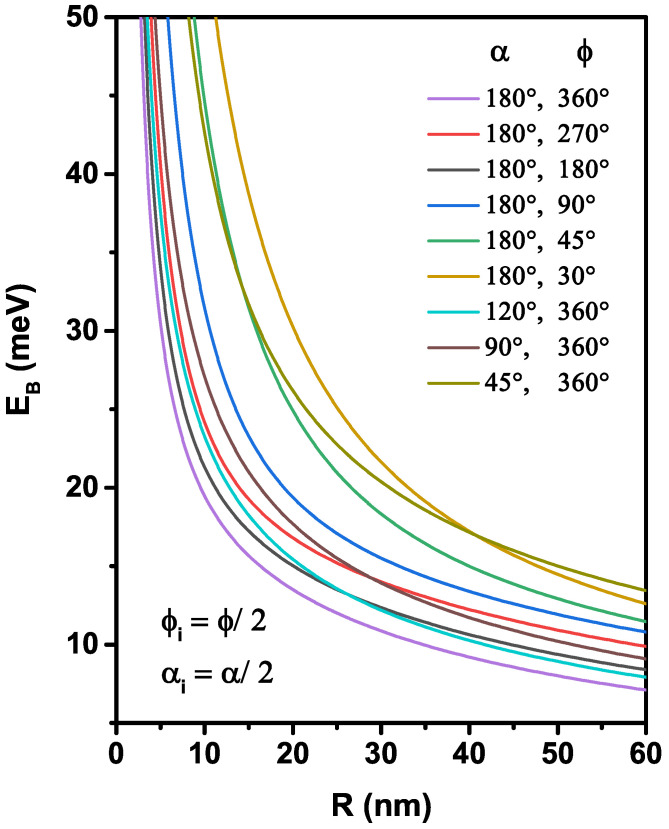
Variation of the binding energy of an on-center donor impurity as a function of nanoflakes radius for nine different setups of the α and ϕ angles.

**Figure 6 nanomaterials-12-00966-f006:**
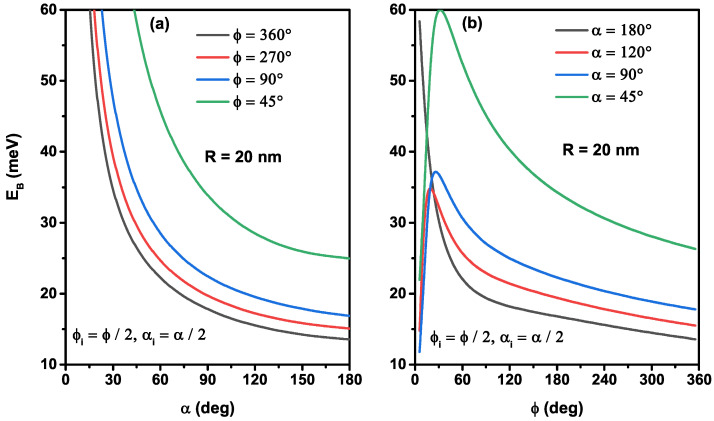
The on-center donor binding energy as a function of the azimuthal angle for several values of the α-angle (**a**) and as a function of the polar angle for several values of the ϕ-angle (**b**). The results are for R=20 nm.

**Figure 7 nanomaterials-12-00966-f007:**
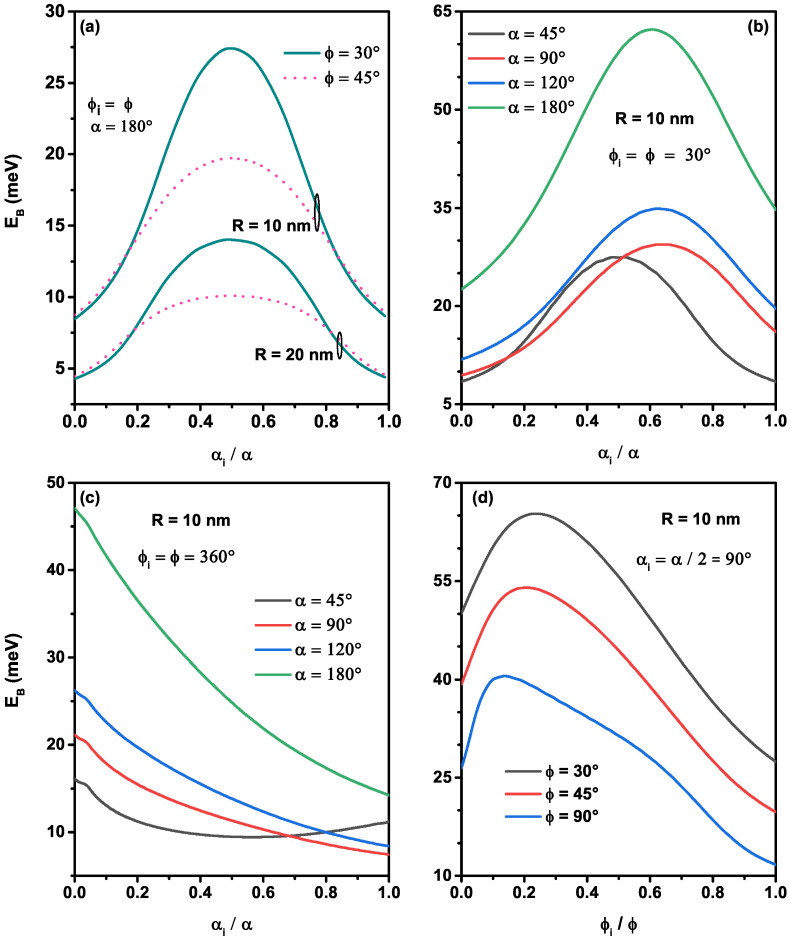
The ground state binding energy as a function of the polar impurity position (**a**–**c**) by considering: two azimuthal angles ($\phi =30^{\circ }$ and $45^{\circ }$) and two nanoflakes radii (R=10 nm and 20 nm) (**a**), different polar angles ($\alpha =45^{\circ },\;90^{\circ },\;120^{\circ }$, and $180^{\circ }$) with $\phi_{i}=\phi =30^{\circ }$ and R=10 nm (**b**), and different polar angles ($\alpha =45^{\circ },\;90^{\circ },\;120^{\circ }$, and $180^{\circ }$) with $\phi_{i}=\phi =360^{\circ }$ and R=10 nm (**c**). In (**d**) the results are a function of the azimuthal impurity position for three azimuthal angles ($\phi =30^{\circ },\;45^{\circ }$ and $90^{\circ }$) with $\alpha_{i}=\alpha =90^{\circ }$ and R=10 nm.

## Data Availability

No new data were created or analyzed in this study. Data sharing is not applicable to this article.

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
