# Peer review of "First Study on the Electronic and Donor Atom Properties of the Ultra-Thin Nanoflakes Quantum Dots"

_nanomaterials, 2022, doi:10.3390/nano12060966_

Round 1
Reviewer 1 Report
In this manuscript, authors investigated the surface properties of an electron confined in spherical ultra-thin quantum dots in the presence of an on-center or off-center donor impurity by considering the Coulombic interaction. The presented work may contribute to a better understanding of the surface state properties of the nanoflakes. This manuscript nevertheless seems to be insufficient, the authors may take the following comments into consideration.
- Under what conditions is the model put forward in this paper? Is it an ideal model?
- Why spherical coordinates are used to obtain ultra-thin 2D nanoflakes with different geometric structures?
- In FIG. 2, how can we draw the conclusion that the electron energy of≤180 and ≤360 tends to the intrinsic energy of the quantum well?
- It is suggested to list the surface value (Snf) of nanoflakes with different radius,α,φ and the same surface structure to further explain and prove "we can get the same nanoflakes surface values (Snf ), but our calculations show that the electron energy in two various structures with the same surface is different".
- “As can be seen from Figure 2a, for some significant values of polar-angle, the donor energy rapidly diminishes as the azimuthal-angle enhances until it reaches a minimum, in which the atom will be more stable, and then increases weakly” should be seen from Figure 4a.
Reviewer 2 Report
In this work, the authors investigated the surface properties of an electron confined in spherical ultra-thin quantum dots in the presence of an on-center or off-center donor impurity by considering the Coulombic interaction. They have developed a novel model which leads to investigate the different nanoflakes geometries by changing the spherical nanoflakes coordinates. This work will contribute to a better understanding of the surface state properties of the nanoflakes. However, I do have some questions, which in my opinion should be addressed.
1) In introduction, the authors write: “As it is known, the electron or hole confinement in 2D-QD or 3D-QD leads to augment the separation between energy levels which produce discrete energy spectrum with the decrease in nanostructures size. Along these lines, the confinement effect has been exploited to improve quantum information transport, sensor performance, and optoelectronic devices, etc [8–11].” The introduction has room to be further improved. Quantum dots (QDs) and quantum wells (QWs) has been used to build advanced components for electronic and opto-electronic applications. It would be great if the authors include these developments and achievements in the introduction, so to give the readers a much broader view. Several recent references concerning on the application of QDs and QWs in optoelectronic devices, such as Applied Physics Letters 97, 011103 (2010); Applied Physics Letters 118, 182102 (2021); Optics Express 27, A669 (2019); etc., should be added, so that the readers can be clear about the state-of-the-art of this topic.
2) The authors write: “…electron effective mass, whith m0 corresponding to the free electron mass.” What is the meaning of “whith”?
3) “Figures 5” should be revised to be “Figure 5”. “Figures 6a” should be revised to be “Figure 6a”. “we have examined in Figures 7” should be revised to be “we have examined in Figure 7”. The authors should check these descriptions throughout the manuscript.
4) The authors writes: “The increase of the confinement effect (the 2D system tend to zero i.e. α → 0 and Ï• → 0) produces a greater electron-donor spatial localization in a small region and their kinetic energy becomes more important.” Here, “the 2D system tend to zero i.e. α → 0 and Ï• → 0” should be revised to be “the 2D system tends to zero i.e. α → 0 and Ï• → 0”.
5) The style of figures should be kept consistent. For example, the EB (meV) of Y-axis should be added in Figures 7a-7d.
6) There are some grammatical errors in the manuscript, although most of them do not obscure the understanding of the technical contents. However, I believe that the paper should be proof-read for English before it is submitted. For example:
--“when the polar angle decrease” should be corrected to be “when the polar angle decreases”.
--“we have found a slight increase of the binding energy” should be corrected to be “We have found a slight increase of the binding energy”
--“we have plotted in Figures 7d…” should be corrected to be “we have plotted in Figure 7d…”
Reviewer 3 Report
The authors presented a theoretical study on Nanoflakes ultra-thin quantum dots. They considered finite difference method to investigate the confined electron in spherical ultra-thin quantum dots in the presence of an on-center or off-center donor impurity by considering the Coulombic interaction.
The study is purely fundamental and the nanoflake geometrical complexity is unusual from a practical point of view. I have some comments that the author should answer/consider ensuring the paper's suitability for publication and therefore the community benefits from the numerical study.
1-What would be the barrier material to be considered for GaAs from a practical point of view to fulfill the infinite barrier approximation?
2- It would be interesting that the authors introduce some sentences in the introduction stating the techniques allowing to fabricate GaAs nanoflake with the considered geometry in this paper.
3. It would be very relevant to provide 2D plots of the electrons probability density (wavefunction) at least for relevant geometries et selected donor impurity position.
4. More practically relevant physical quantities should be provided and discussed such as Oscillator strength and absorption coefficient
Round 2
Reviewer 3 Report
The authors performed most of the recommended changes
the paper can be accepted for publication